# The Properties of Intermediate-Temperature Solid Oxide Fuel Cells with Thin Film Gadolinium-Doped Ceria Electrolyte

**DOI:** 10.3390/membranes12090896

**Published:** 2022-09-17

**Authors:** Andrey Solovyev, Anna Shipilova, Egor Smolyanskiy, Sergey Rabotkin, Vyacheslav Semenov

**Affiliations:** 1Institute of High Current Electronics SB RAS, 634055 Tomsk, Russia; 2Laboratory of Pulse-Beam, Electric Discharge and Plasma Technologies, Tomsk Polytechnic University, 30, Lenin Prospekt, 634050 Tomsk, Russia

**Keywords:** solid oxide fuel cells, electrolyte thin film, mixed ionic-electronic conductor, gadolinium-doped ceria, magnetron sputtering

## Abstract

Mixed ionic-electronic conducting materials are not used as a single-layer electrolyte of solid oxide fuel cells (SOFCs) at relatively high operating temperatures of ~800 °C. This is because of a significant decrease in the open-circuit voltage (OCV) and, consequently, the SOFC power density. The paper presents a comparative analysis of the anode-supported SOFC properties obtained within the temperature range of 600 to 800 °C with yttria-stabilized zirconia (YSZ) electrolyte and gadolinium-doped ceria (GDC) electrolyte thin films. Electrolyte layers that are 3 µm thick are obtained by magnetron sputtering. It is shown that at 800 °C, the SOFC with the GDC electrolyte thin film provides an OCV over 0.9 V and power density of 2 W/cm^2^. The latter is comparable to the power density of SOFCs with the YSZ electrolyte, which is a purely ionic conductor. The GDC electrolyte manifests the high performance, despite the SOFC power density loss induced by electronic conductivity of the former, which, in turn, is compensated by its other positive properties.

## 1. Introduction

The power density improvement of solid oxide fuel cells (SOFCs) is important for their commercial attractiveness. The higher the fuel cell performance, the smaller the SOFC power generation system; less material is required, providing its lower cost. The area-specific resistance of SOFCs is mostly composed of the polarization resistance and ohmic resistance. Since the ohmic resistance is largely associated with the electrolyte, two approaches can be used to improve the SOFC performance: the electrolyte thickness reduction and the use of the electrolyte material with an ionic conductivity higher than that of the conventional yttria-stabilized zirconia (YSZ) electrolyte. The electrolyte thickness can be successively reduced to a few micrometers by using physical vapor deposition (PVD), such as pulsed laser deposition [1], and magnetron sputtering [2]. PVD methods mitigate the problems associated with high-temperature sintering (>1200 °C), which are characteristic of traditional ceramic methods and only need annealing at relatively low temperatures. The second approach is implemented by using electrolytes based on bismuth oxide in its δ-phase (high-temperature fluorite structure) [3], strontium-and-magnesium-doped lanthanum gallate [4], proton conductors (doped BaCeO_3_) [5], or samarium- or gadolinium-doped ceria [6]. 

In this work, these two approaches are used to improve the properties of anode-supported SOFCs in the temperature range of 600–800 °C, i.e., SOFCs with the gadolinium-doped ceria (GDC, Ce_0.8_Gd_0.2_O_1.9−x_) electrolyte thin film obtained by magnetron sputtering. The GDC electrolyte is used for its ionic conductivity, which, in the temperature range of 600–800 °C, is 4 or 5 times higher than that of the commonly used YSZ electrolyte [7]. Moreover, CeO_2_-based materials are thermodynamically stable in the presence of water and hydrocarbon vapors and chemically compatible with cobalt-containing perovskite oxide cathodes [8]. 

It should be noted that screen-printed GDC electrolytes are not well densified even at 1400 °C sintering [9]. Different sintering aids are studied to densify the doped ceria electrolyte, but it is still a problem to lower the sintering temperature below 1250 °C [10,11]. Thus, the PVD technology is one of the best ways to obtain doped ceria thin films.

Nevertheless, the use of the GDC electrolyte thin film as a single-layer GDC electrolyte is complicated by a partial Ce^4+^ reduction to Ce^3+^ at a low partial pressure of oxygen. This provides electronic conductivity in the material and, consequently, the reduction in both the open-circuit voltage (OCV) and SOFC power density [12]. Therefore, the single-layer GDC electrolyte is mostly used to create low-temperature solid oxide fuel cells, and its properties are investigated at temperatures not exceeding 650 °C [13,14,15,16,17,18].

Nevertheless, the electronic conductivity impact on the GDC electrolyte SOFC performance is very sensitive to the fuel utilization efficiency. The SOFC performance grows with increasing fuel utilization efficiency due to the rise in the oxygen partial pressure, and electronic conductivity reduces with the increasing oxygen partial pressure [19].

At the same time, Pérez-Coll et al. [20] showed that some minor electronic conductivity of the electrolyte could be beneficial for the improvement of electrochemical processes on electrodes due to the expansion of the electrochemical reaction zone. Thus, electronic conductivity of the electrolyte could reduce, to some extent, the SOFC polarization resistance. 

There is virtually no information offered in the literature concerning the SOFC properties with the single-layer GDC electrolyte thin film within the temperature range of 600 to 800 °C. In our earlier research [21], we show the excellent power output with a peak power density as high as 1.07 W/cm^2^ at 750 °C, despite relatively low OCV values (0.77–0.8 V) for the anode-supported SOFC with the GDC electrolyte. Still, the properties of SOFCs with a single-layer electrolyte of purely ionic conductivity have never been compared to SOFCs with an electrolyte possessing mixed ionic-electronic conductivity.

The aim of this work is to study the SOFC performance with a single-layer GDC electrolyte at high operating temperatures (up to 800 °C). Current voltage and power characteristics are measured for anode-supported SOFCs with a GDC electrolyte, which are then compared with the properties of SOFCs with the thin film YSZ electrolyte.

## 2. Materials and Methods

GDC and YSZ electrolyte layers, 3 μm thick, were deposited by reactive magnetron sputtering of 30 × 10 cm^2^ targets made of Ce-Gd (90:10 at.%) and Zr-Y (85:15 at.%) [22] onto the NiO/10ScCeSZ anodes (Kceracell Co., Chungcheongnam-Do, Korea) with a diameter of 20 mm and thickness of 0.5 mm. Prior to the electrolyte deposition, the anode substrates were treated in a vacuum chamber using the anode layer ion source. The 10 min treatment was performed at a 2 kV discharge voltage and 100 mA discharge current. The electrolyte was deposited onto the substrate heated up to 350 °C, using a 45 kHz pulse frequency and 4 kW discharge power for the Zr-Y target (Girmet, Moscow, Russia) and 3 kW discharge power for the Ce-Gd target (Girmet, Moscow, Russia). The deposition rate was 0.5 and 0.7 μm/h for the YSZ and GDC layers, respectively. The obtained specimens were then annealed at 1200 °C for 1 h to improve the electrolyte crystallinity. The La_0.6_Sr_0.4_CoO_3_ (LSC) cathode (Kceracell Co., Chungcheongnam-Do, Korea), 1 cm^2^ in area, was screen-printed on the electrolyte film surface. Cathode sintering was performed at 800 °C during the fuel cell electrochemical test. 

Electrochemical investigations were performed in the temperature range of 600 to 800 °C at the constant supply of dry hydrogen (120 mL/min) and air (350 mL/min) to the anode and cathode, respectively. The electrochemical performance was studied on a PL-150 potentiostat and a Z-500P impedance spectrometer (Elins, Chernogolovka, Russia). 

A VERSA 3D HighVac (FEI, Hillsboro, OR, USA) system, combining a scanning electron microscope (SEM) and focused ion beam microscope, was used to study the microstructure of the fabricated fuel cells at a 5 kV accelerating voltage.

## 3. Results

Figure 1 presents SEM images of cross sections and the surface of the GDC and YSZ electrolyte deposited onto the anode surface. The images were made after the cell testing. In both cases, the electrolyte structure is rather dense, and its adhesion to the anode is high. Both electrolytes are crack-free and have no apparent columnar structure.

Current voltage characteristics of the SOFC with the GDC electrolyte are plotted in Figure 2a. The open-circuit voltage of the SOFC with the GDC electrolyte decreases from 0.97 to 0.93 V with the temperature growth from 600 to 800 °C. The maximum power density is 560, 1375, and 2250 mW/cm^2^ at 600, 700, and 800 °C, respectively.

OCV values obtained for the GDC electrolyte thin film were the highest as compared to the literature data and close to the theoretical value of 0.98 V obtained at 600 °C in [23]. This indicated that the dense GDC electrolyte thin film correlated with the SEM observations. Duncan et al. [24] reported on the 0.93 V open-circuit voltage measured at a lower temperature of 600 °C and a significantly higher thickness of 30 µm for the GDC electrolyte. 

The obtained power density is higher than that described in [15,17]; namely, its maximum values are 370 and 380 mW/cm^2^ at 600 °C for SOFCs with 3.3 and 10 µm-thick GDC electrolytes fabricated by aerosol pyrolysis and tape casting, respectively. This indicates the high electrical performance obtained both at low and high operating temperatures for the anode-supported SOFC with the magnetron-sputtered GDC electrolyte.

Using the NiO-GDC anode, 15 µm-thick GDC electrolyte, and LSCF-GDC composite as a cathode material, Fu et al. [25] reported on the higher SOFC performance, namely the maximum power density of 625 and 909 mW/cm^2^ obtained respectively at 600 and 650 °C. Unfortunately, experiments were not conducted at higher temperatures, and the high SOFC performance was gained due to the presence of the GDC in the composite anode and cathode. 

Depending on the temperature, the open-circuit voltage of the SOFC with the YSZ electrolyte ranges between 1.08 and 1.1 V (Figure 2b). The maximum power density is 235, 950, and 2300 mW/cm^2^ at 600, 700, and 800 °C, respectively. Therefore, at 800 °C, both fuel cells have comparable power density. However, with the decreasing temperature, the power density of the SOFC with the GDC electrolyte becomes higher than that of the SOFC with the YSZ electrolyte. The lower the operating temperature, the bigger advantage the SOFC has with the GDC electrolyte. At 600 °C, the power density of the SOFC with a single-layer GDC electrolyte is twice as much as that of the SOFC with the YSZ electrolyte.

In Figure 3a, impedance spectra belong to the SOFC with the GDC electrolyte measured under the open-circuit conditions at 600–800 °C. Figure 3b shows the respective imaginary impedance plots. One can see the Nyquist plots with two separated arcs as it is commonly observed for the SOFC with Ni/YSZ anodes. The ohmic *R*_Ω_ and faradaic *R*_F_ impedances are determined respectively by the first intercept and the difference between the first and second intercepts on the real axis. At 600, 700, and 800 °C, the ohmic impedance *R*_Ω_ is 0.24, 0.1, and 0.06 Ohm cm^2^, whereas the faradaic impedance *R*_F_ is 1.13, 0.2, and 0.03 Ohm cm^2^, respectively. The total cell impedance representing the sum (*R*_Ω_ + *R*_F_) of the ohmic and faradaic impedances lowers with the increasing temperature, i.e., the ohmic impedance reduces by 4 times, while the faradaic impedance reduces by 38 times. At 800 °C, the ohmic impedance contributes greatly to the total cell impedance and includes the electrolyte resistance, electrode ohmic resistance, and contact resistance between the electrodes and electrolyte. At 600 °C, the SOFC impedance is mostly determined by the faradaic impedance, depending on the electrode processes, such as gas diffusion and conversion and charge transfer reactions on the anode and cathode.

According to Figure 3b, at 800 °C, the high-frequency peak is observed at ~1000 Hz, while the low-frequency peak occurs at ~2 Hz. When the temperature lowers, the high-frequency peak intensity grows and shifts toward the lower frequency region due to higher resistance. This behavior means thermal activation and can be interpreted as physicochemical electrode processes [26]. The low-frequency peak intensity grows at temperature below 700 °C, which indicates a less dependent temperature behavior.

At temperatures up to 800 °C, the power density of the SOFC with the GDC electrolyte is higher than that with the YSZ electrolyte, despite GDC electronic conductivity. The first reason is higher conductivity of the GDC electrolyte against that of the YSZ electrolyte. The second is the expansion of the electrochemical reaction zone, according to Pérez-Coll et al. [20]. Electronic conductivity of an electrolyte promotes the electronic transport, thereby enhancing the oxygen exchange on the surface [27,28]. Third, improved ionic conductivity results in the oxygen transport rate increase at the electrolyte-electrode interface, which reduces the electrode overpotential [29,30] and allows the improvement of the cell performance also. Uchida et al. [29] report that the exchange current density grows as a square of ionic conductivity for all electrolytes operating within 800–1000 °C. Thus, the oxygen transport rate at the electrolyte-electrode interface increased by using an electrolyte with higher ionic conductivity and reduced the anodic overpotential. All abovementioned reasons for a higher power density compensate the power loss caused by the OCV reduction for the SOFC with the GDC electrolyte.

The power density of the SOFC with the thin GDC electrolyte is compared with the SOFC performance within 700–800 °C reported in [31,32,33,34], which further illustrates its excellent performance (see Table 1).

Thus, the advantages of a single-layer GDC electrolyte over a single-layer YSZ electrolyte are recognized not only at low temperatures (600–650 °C) but also at relatively high operating temperatures of ~800 °C. For example, in using the cobalt-containing oxide cathode in the range of 750 to 800 °C, a single-layer YSZ electrolyte requires the formation of a barrier layer near the cathode. This complicates the SOFC production process and makes it more expensive.

Future work will involve the long-term stability testing and characterization and research into the resistance to thermal and redox cycling of single cells with the GDC electrolyte.

## 4. Conclusions

In this paper, the properties of the anode-supported SOFC with magnetron-sputtered electrolyte thin films that have ionic (YSZ) and mixed ionic-electronic (GDC) conductivities were compared within the range of 600–800 °C. The fuel cell with the GDC electrolyte with a thickness of 3 µm manifested the high electrical performance both at low and high operating temperatures. An open-circuit voltage of 0.97 and 0.93 V was achieved at 600 and 800 °C, respectively. The maximum power density was found to be 560, 1375, and 2250 mW/cm^2^ at 600, 700, and 800 °C, respectively. For the first time, it was shown that at a high operating temperature of 800 °C, the SOFC with the single-layer GDC electrolyte possessed high OCV and power density comparable to the SOFC with the YSZ electrolyte. That was explained by the GDC electrolyte conductivity, which was higher than that of the YSZ electrolyte, resulting in the increase in the oxygen transport rate and the electrode overpotential reduction for the SOFC with the GDC electrolyte.

## Figures and Tables

**Figure 1 membranes-12-00896-f001:**
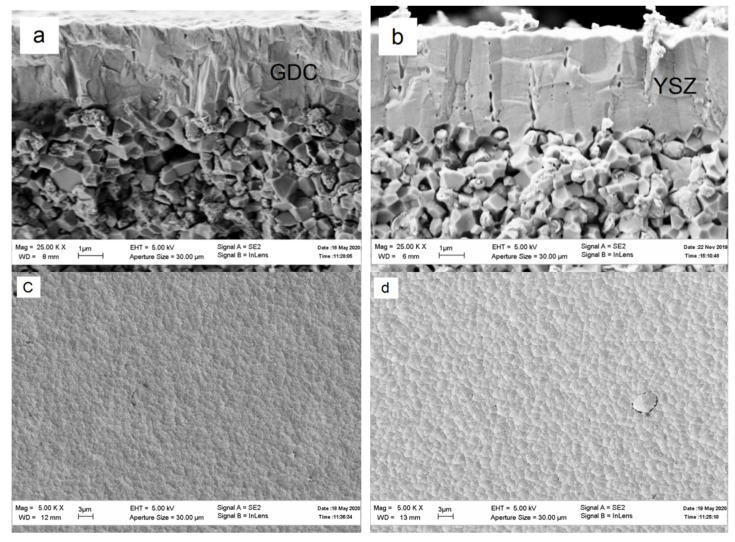
SEM images of the cross sections and surface of the GDC (**a,c**) and YSZ (**b,d**) electrolyte microstructure.

**Figure 2 membranes-12-00896-f002:**
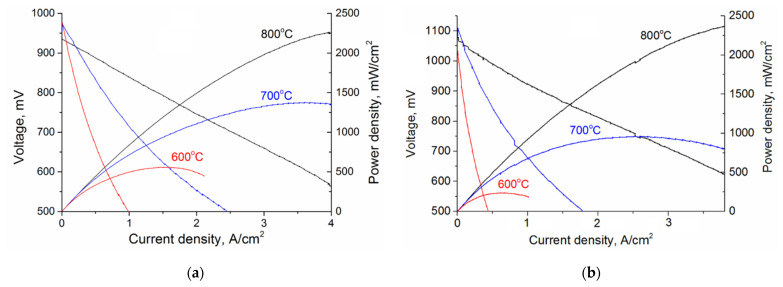
I-V curves of SOFC with GDC (**a**) and YSZ (**b**) electrolyte thin films.

**Figure 3 membranes-12-00896-f003:**
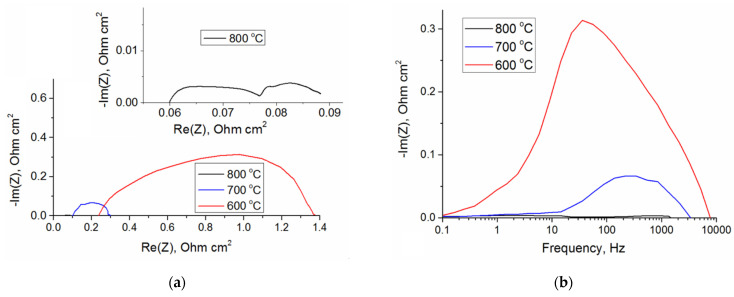
Impedance spectra of the SOFC with the GDC electrolyte (measured at 600–800 °C with open-circuit voltage) (**a**); Bode plots of the imaginary impedance (**b**).

**Table 1 membranes-12-00896-t001:** Power density values of intermediate-temperature SOFCs with the GDC electrolyte.

Cell Configuration	Power Density, mW/cm^−2^	Temperature, °C	Reference
NiO-GDC|GDC 180 μm|BSFC	510	700	[31]
NiO-GDC|GDC 300 μm|PBFN	520	700	[32]
NiO-GDC|GDC 320 μm|PNO-PCO	570	800	[33]
NiO-GDC|GDC 250 μm|PSFN	699	800	[34]
NiO/10ScCeSZ |GDC 3 μm|LSC	1375	700	This work

## Data Availability

The data presented in this study are available on request from the corresponding author.

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
