# Peer review of "The Properties of Intermediate-Temperature Solid Oxide Fuel Cells with Thin Film Gadolinium-Doped Ceria Electrolyte"

_membranes, 2022, doi:10.3390/membranes12090896_

Round 1

Reviewer 1 Report

This manuscript reports a comparion of Ni-ScCeSZ anode-supported solid oxide fuel cells based upon GDC and YSZ electrolyte with a thickness of 3 um by magnetron sputtering processing. Interestingly, the single cell with GDC electrolyte obtained a high OCV of over 0.9 V and a good power density of 2 W/cm2 at 800℃. This paper can be accepted after some revision as follows:

1. Was the high OCV of over 0.9 V in the as-prepared single cell with GDC electrolyte stable for long time?

2. Please give the SEM mages for the surface of electrolytes in the revision.

3. The long-term stability of the as-prepared single cell with GDC electrolyte is needed for this paper.

4. Is GDC better than YSZ? Why and how? Thank you.

Author Response

Responses to Reviewer #1’s comments:

  1. COMMENT: Was the high OCV of over 0.9 V in the as-prepared single cell with GDC electrolyte stable for long time?

RESPONSE: The OCV was over 0.9 V for 4 hours when the volt-current characteristics and impedance spectra were measured. Unfortunately, longer tests were not carried out.

  1. COMMENT: Please give the SEM mages for the surface of electrolytes in the revision.

RESPONSE: SEM images of the electrolytes surface were added in the revised paper.

  1. COMMENT: The long-term stability of the as-prepared single cell with GDC electrolyte is needed for this paper.

RESPONSE: Unfortunately, the long-term stability of the single cell with GDC electrolyte was not investigated in this work.

  1. COMMENT: Is GDC better than YSZ? Why and how? Thank you.

RESPONSE: This question cannot be answered definitively, because each of the electrolytes has its own advantages and disadvantages. And in general it is better to use them together in a two or three-layer structure, as many do. YSZ layer blocks electronic conductivity, and GDC layers are better compatible with cathode materials or GDC-containing anode.

If we mean fuel cells with single-layer thin electrolyte of anode- or metal-supported construction, GDC electrolyte looks preferable to YSZ at temperatures less than 800C. At least in terms of power density. Besides, the single-layer YSZ electrolyte requires the formation of a barrier layer near the cathode. This complicates and the SOFC production process and makes it more expensive. However, long-term stability, resistance to redox cycling and thermocycling need to be investigated further.

Literature data show that the stability of SOFCs with GDC electrolyte has not been studied as intensively as with YSZ electrolyte. Nevertheless, there are several works in this area. Ko et al. [Structural Stability of the GDC Electrolyte for Low Temperature SOFCs Depending on Fuels, Electrochemical and Solid-State Letters, 13(10) B113-B115, 2010] showed that dry methane fuel did not cause any bad effects on the structural stability of the GDC electrolyte. The GDC electrolyte supported cell with the Cu/GDC–GDC anode showed the maximum power density of 0.1 Wcm–2 and long-term stability for more than 500 h at 650 °C in dry methane atmosphere [https://doi.org/10.1002/fuce.200800186].

Lee et al. [Tailoring gadolinium-doped ceria-based solid oxide fuel cells to achieve 2 Wcm-2 at 550°C, NATURE COMMUNICATIONS, 2014, 5:4045, DOI: 10.1038/ncomms5045] showed that stable and high performance at low temperatures can be obtained using the GDC-based LT-SOFC in a configuration of the BSCF-GDC cathode | GDC electrolyte (5 mm) | nanocomposite Ni-GDC AFL | Ni-GDC anode (the voltage degradation was approximately 5.6% over 250 h with a degradation rate of 0.00017 Vh-1.).

Meng et al. [Heterointerface Effect in Accelerating the Cathodic Oxygen Reduction for Intermediate-Temperature Solid Oxide Fuel Cells. Front. Chem. (2022) 10:959863. doi: 10.3389/fchem.2022.959863] studied the stability of cell with 250-µm-thick GDC electrolyte. A 100-h long-term stability test was conducted under the conditions of 0.3 A cm-2 and 700°C. The initial voltage of Cell decreased from 0.738 to 0.712 V after 100 h of polarization, and the degradation rate of OCV was about 0.0352% h-1.

Reviewer 2 Report

The authors prepared and characterized the morphology and electrochemical properties of Thin Film Gadolinium-Doped Ceria Electrolyte for Intermediate-Temperature Solid Oxide Fuel Cells application.

I consider the content of this manuscript will meet the reading interests of the readers of the Membranes journal. The authors need to clarify some issues or supply some more experimental data to enrich the content. This could be comprehensive and meaningful work after revision.

1. For grammar issues, I suggest the authors double-check the minor grammar errors in the manuscript, especially the lack of and redundant use of definite articles and incorrect prepositions.

2. (Line 168-169)The first reason is the higher conductivity of the GDC electrolyte as against that one with the YSZ electrolyte.’

What is the conductivity of the GDC and YSZ electrolytes? Authors could calculate it from the impedance data obtained.

3. The authors are highly recommended to perform other characterizations (e.g., structural, mechanical, and thermal properties) of the electrolyte to acquire comprehensive results. This is probably for future work since a communication article usually only presents preliminary results or significant findings.

Author Response

Responses to Reviewer #2’s comments:

  1. COMMENT: For grammar issues, I suggest the authors double-check the minor grammar errors in the manuscript, especially the lack of and redundant use of definite articles and incorrect prepositions.

RESPONSE: English has been checked again, changes are highlighted.

  1. COMMENT: (Line 168-169) ‘The first reason is the higher conductivity of the GDC electrolyte as against that one with the YSZ electrolyte.’ What is the conductivity of the GDC and YSZ electrolytes? Authors could calculate it from the impedance data obtained.

RESPONSE: In this sentence we meant that GDC electrolyte has ionic conductivity 4 or 5 times higher than that of the YSZ electrolyte in the temperature range of 600–800 °С [DOI: 10.1016/S0167-2738(99)00318-5]. From the impedance data it is difficult to determine the ionic conductivity of the thin-film electrolyte. It is possible to determine the Ohmic resistance of the whole cell. But it includes the electrolyte resistance, electrode ohmic resistance, contact resistance between electrodes and electrolyte.

  1. COMMENT: The authors are highly recommended to perform other characterizations (e.g., structural, mechanical, and thermal properties) of the electrolyte to acquire comprehensive results. This is probably for future work since a communication article usually only presents preliminary results or significant findings.

RESPONSE: Yes, that was not the purpose of this paper. Nevertheless, some results of the study of GDC electrolyte formed by magnetron sputtering are presented in papers [21, 22] referenced in this paper.